# Community Pharmacist Consultation Service: A Survey Exploring Factors Facilitating or Hindering Community Pharmacists’ Ability to Apply Learnt Skills in Practice

**DOI:** 10.3390/pharmacy10050117

**Published:** 2022-09-21

**Authors:** Elizabeth M. Seston, Chiamaka Julia Anoliefo, Jinghua Guo, Joanne Lane, Chikwado Okoro Aroh, Samantha White, Ellen I. Schafheutle

**Affiliations:** 1Centre for Pharmacy Workforce Studies (CPWS), Division of Pharmacy & Optometry, Faculty of Biology, Medicine and Health, The University of Manchester, Manchester M13 9PT, UK; 2Centre for Pharmacy Postgraduate Education (CPPE), Division of Pharmacy & Optometry, Faculty of Biology, Medicine and Health, The University of Manchester, Manchester M13 9PT, UK

**Keywords:** community pharmacy, clinical skills, urgent care, minor ailments

## Abstract

Background: The NHS Community Pharmacist Consultation Service (CPCS) offers patients requiring urgent care a consultation with a community pharmacist, following referral from general practice or urgent care. The study explored the impact of undertaking a Centre for Pharmacy Postgraduate Education (CPPE) CPCS learning programme, and barriers and enablers to CPCS delivery. Methods: CPPE distributed an online survey to those who had undertaken their CPCS learning. The survey explored participants’ knowledge, confidence and application of taught skills/tools, including clinical history-taking, clinical assessment, record keeping, transfer of care, and Calgary-Cambridge, L(ICE)F and SBARD communication tools. Details on barriers and enablers to CPCS delivery were also included. Results: One-hundred-and-fifty-nine responses were received (response rate 5.6%). Knowledge of, and confidence in, taught skills were high and respondents reported applying skills in CPCS consultations and wider practice. Barriers to CPCS included a lack of general practice referrals, staffing levels, workload, and GP attitudes. Enablers included a clear understanding of what was expected, minimal concerns over indemnity cover and privacy, and positive patient attitudes towards pharmacy. Conclusion: This study demonstrates that community pharmacists can extend their practice and contribute to the enhanced provision of urgent care in England. This study identified barriers, both interpersonal and infrastructural, that may hinder service implementation.

## 1. Introduction

It has previously been recognised that in England, some patients are attending primary care services in the National Health Service (NHS), such as general practice and emergency departments, with what have been described as ‘clinically divertible’ urgent, low-acuity conditions [1]. The diversion of patients with urgent, low-acuity conditions to community pharmacy has been calculated as potentially saving the NHS over GBP 1 billion per annum [2]. In recognition of this, and the fact that community pharmacies are ideally placed to provide cost-effective care for such patients [3], the NHS Community Pharmacist Consultation Service (CPCS) was launched in October 2019 [4]. The CPCS is a national advanced service designed to foster the integration of community pharmacy into local NHS urgent primary care services [5], offering patients requiring urgent care a face-to-face or telephone consultation with a pharmacist, following referral from NHS 111, general practice, and other urgent care services. The aim is to provide convenient and accessible treatment closer to patients’ homes and to relieve pressure on general practice and emergency departments. The establishment of the CPCS was informed by previous services, particularly the digital minor illness referral scheme [6].

The CPCS is part of the wider policy drive for pharmacists to make better use of their clinical skills, and for community pharmacies to become a more integrated part of primary care [7]. It is important to ensure pharmacists have the skills to respond appropriately to patients presenting with an extended range of urgent, low-acuity conditions. NHS England (NHSE), through the Pharmacy Integration Fund (PhIF), therefore commissioned the Centre for Pharmacy Postgraduate Education (CPPE), the Health Education England (HEE) funded national provider for post-registration pharmacy education in England, to develop appropriate learning resources. The CPPE learning aimed to equip pharmacists with the skills required for autonomous, person-centred consultations, where the focus (and indeed payment) is on advice rather than the supply of an over-the-counter (OTC) product.

There is evidence that community pharmacists are motivated to take part in extended services, and a number of facilitators for clinical service implementation have been identified, including staffing levels with appropriate skill mix, good consultation skills, and provision of training [6,8]. Nonetheless, a number of barriers to the implementation of community pharmacy services have been identified, including workload pressure, a lack of suitable privacy for consultations, a lack of patient awareness of community pharmacy services, a lack of confidence in delivering clinical services, a lack of clear service specification, a lack of general practitioner (GP) awareness, and negative attitudes towards community pharmacy services [5,8,9,10,11,12,13].

Whilst there is evidence to indicate that taking part in training programmes for new services can enhance confidence and impact on behaviour in practice, it has been suggested that the acquisition of skills alone is not sufficient for behaviour change due to limited opportunities to apply skills, particularly in community pharmacy [5,14,15]. The initial uptake of the CPCS has also been low, with community pharmacies receiving just 7000 CPCS referrals per week [16], which is low in light of 11,600 community pharmacies in England. This has led to calls to improve the CPCS referral process, particularly from general practice [17]. Indeed, there are plans to extend referral, and a pilot is currently ongoing which allows referrals from emergency departments (EDs) and urgent treatment centres (UTCs) [18]. It is therefore important to explore the impact of learning programmes in practice, and to understand barriers and enablers to the implementation of new services in practice.

The overall aims of the research were to explore the impact of undertaking the CPPE CPCS learning on the delivery of the CPCS and other consultations, and to identify barriers and enablers to CPCS.

The objectives were:to explore whether pharmacists who had completed the CPPE CPCS training had delivered the CPCS to deal with referrals from NHS 111, general practice and other services;to explore participants’ knowledge of, and confidence in, specific elements of the CPPE CPCS learning, and whether they were applying these aspects for CPCS and/or other community pharmacy consultations, such as for minor ailments, new medicines, or public health services;to explore barriers and enablers to providing the NHS CPCS service;to explore differences according to the pharmacist characteristics and pharmacy type.

## 2. Materials and Methods

The CPCS learning programme developed by CPPE involved preparatory online learning, an assessment of learning needs, and a workshop. The learning introduced pharmacists to a range of tools and techniques to underpin clinical, person-centred practice. Workshops focused on enhancing pharmacists’ communication and clinical and physical examination skills, thus equipping them with the necessary skills to resolve low-acuity conditions.

The CPPE CPCS workshops were delivered face-to-face between October 2019 and March 2020, pivoting to online delivery (due to COVID-19) from October 2020 to September 2021. Some changes were made when moving to full online delivery, with some content previously covered during face-to-face workshops added to the preparatory e-learning. The total for all learning is calculated to be about 16 h (including self-assessment). Prior to the pandemic, the workshop was 7 h in duration with about 9 h of pre-workshop learning; since moving fully online, the self-directed e-course takes 10–12 h and the online workshop takes 3:15 h. All workshops were facilitated by clinicians.

A cross-sectional questionnaire was distributed to pharmacists who had taken part in the CPCS training and who consented to being contacted by CPPE. Pharmacist views were collected at least 6 weeks after they had completed the training. The questionnaire was designed in consultation with CPPE staff involved in CPCS learning design and informed by existing research. Questions on knowledge, confidence and skill use in practice were informed by the CPPE CPCS intended learning outcomes (ILOs) and intended changes in the pharmacists’ behaviours.

The questionnaire was divided into four sections. Section 1 asked respondents if the pharmacy they worked in offered the CPCS, the number of referrals they had received from NHS 111, general practice (or other sources), and the type of learning they had undertaken (face-to-face workshops vs. fully online), and their main reason for undertaking the learning.

Section 2 of the questionnaire explored the participants’ learning of the relevant skills covered in the CPPE CPCS training: Calgary-Cambridge, L(ICE)F (ICE (Ideas, Concerns and Expectations) is a concept in consultation skills training to **elicit the patient’s agenda**. It has been shown to reduce overdiagnosis and over-treatment. ICE has subsequently been updated to L(ICE)F to include the lifestyle and feelings (lifestyle, ideas, concerns, expectations, feelings)), SBARD (SBARD stands for Situation, Background, Assessment, Recommendation, Decision and is a tool that enables information to be transferred accurately between individuals.), clinical history-taking, clinical assessment, clinical record-keeping, and clinical transfer of care. Participants were asked to rank their level of agreement (on a 5-point Likert-type scale, where 1 = strongly disagree and 5 = strongly agree) with various statements relating to their:knowledge of those skills;confidence in using those skills;application of those skills in the delivery of CPCS and in wider practice (participants were not asked the questions about whether they used the SBARD tool or enacted a warm transfer of care during CPCS or other community pharmacy consultations).

Section 3 included a series of statements related to perceived enablers and barriers to CPCS provision, including the extent and appropriateness of referrals, staffing and workload issues, privacy and indemnity concerns, and relationships with and integration into the primary healthcare team. Statements were informed by previous research on barriers and enablers to the delivery of new community pharmacy services [6,14].

Respondents were also asked to rate their self-reported confidence and competence in delivering the service on a 10-point scale, where 1 = not at all confident/competent, 10 = very confident/competent. The final section of the questionnaire gathered background information about the participants themselves (year of registration, type of community pharmacy where they worked most commonly, role within the pharmacy, gender, age).

The questionnaire was created on the OnlineSurveys platform, piloted, and sent by CPPE via an email containing a survey link to 2836 pharmacists who attended CPCS training between October 2019 and October 2021 and had given consent to be contacted by CPPE for marketing and evaluation purposes. The survey was distributed in November 2021 and kept open for two months; one reminder was sent in December 2021.

### 2.1. Ethical Approval

Prior to conducting the study, the University of Manchester Ethics Decision Tool confirmed that ethical approval was not required due to the questionnaire being anonymous, the data collected not being considered sensitive or confidential, participants not being from vulnerable or dependent groups, and there being no risks of participants disclosing illegal/unprofessional conduct.

### 2.2. Data Handling and Analysis

Data were downloaded from the OnlineSurveys platform, cleaned where necessary and uploaded into SPSS v.27 (IBM). Basic descriptive analysis was completed, with the number (*n*) and percentages reported for categorical variables. Inferential statistics (χ^2^) were used to compare variables across different subgroups, where relevant, with a significance level set at 5%, meaning that one can be confident that any significant result has not occurred by chance alone. Independent samples *t*-tests were used to compare mean values across subgroups where relevant. Spearman’s rank was used to measure correlation between variables. Only significant findings are noted in the results.

Some variables were recoded for subgroup analysis. These included the pharmacist’s role (locum vs. employed pharmacist/manager), pharmacy type (small/medium/multiple pharmacies vs. independent pharmacies) and year of qualification (pre-2000 vs. 2000 and later). The latter was chosen as those who registered post-2000 are more likely to have been impacted by changes to the initial education and training of pharmacists. For the knowledge, skills, application, barrier and enabler statements, the agreement responses were recoded into binary variables (agree/neither agree vs. disagree). 

## 3. Results

### 3.1. Response Rate and Respondent Characteristics

One-hundred-and-fifty-nine survey responses were received (response rate 5.6%). Sixty-six percent of respondents were female, the most populous age group was 55–64 years (a third of all respondents), and the most common registration years were 1980–1989 (30.2%). For further details on respondents’ characteristics, see Table 1.

### 3.2. Employment Status and Type of Community Pharmacy

Forty-three percent (*n* = 68) of respondents were working in a large multiple community pharmacy, 33.3% (*n* = 53) in an independent pharmacy, and 17.0% (*n* = 27) in a small-to-medium pharmacy. Eleven respondents (6.9%) had not worked in a community pharmacy in the previous 12 months. Fifty-five percent of respondents were employed pharmacists (*n* = 88); 32 of these were pharmacist managers. Locum pharmacists represented 38.4% (*n* = 61) of all respondents, and 6.3% (*n* = 10) were working in another role.

### 3.3. Provision of the CPCS and Number of Referrals

One-hundred-and-forty-two (of 159) respondents (89.2%) were currently working in a community pharmacy that was offering the CPCS for minor illness consultations. Respondents received a total of 399 CPCS referrals from NHS 111, 357 from general practice and 4 from ‘other’ sources in the four weeks prior to completing the survey. Respondents received a mean of 2.63 (SD ± 3.363) referrals from NHS 111 and a mean of 2.35 (SD ± 7.25) referrals from general practice. Twenty-eight percent of all respondents (*n* = 45) had not received a referral from NHS 111 in the past 4 weeks and 53% (*n* = 84) had not received a referral from general practice. Overall, 22% of respondents (*n* = 35) had not received a CPCS referral from either NHS 111 or general practice in the previous 4 weeks. There were no significant differences in the number of either NHS 111 or general-practice referrals by type of pharmacy (multiple vs. independent).

### 3.4. Type of Training and Reasons for Engaging with the Training

Three quarters (*n* = 119) of respondents had attended the CPPE CPCS workshop face-to-face, and 40 (25%) had done so online. The main reason for engaging with the CPCS training was to be able to deliver the CPCS (66%, *n* = 105). Eighteen percent of respondents (*n* = 29) engaged with the training to increase their confidence in minor ailment consultations in general. Other reasons included ‘as part of CPD’ (9%, *n* = 12), ‘to develop clinical knowledge on minor ailments’ (6%, *n* = 9) and ‘other’ (other responses included the training counting towards foundation pharmacist training pathway and to provide support for other pharmacists providing the service) (2%, *n* = 4).

### 3.5. Self-Reported Competence and Confidence in Delivering the CPCS

The mean self-reported competence and confidence were 7.9 ± 1.494 (SD) and 7.9 ± 1.581 (SD), respectively, indicating that respondents generally felt competent and confident in delivering the CPCS. There was a strong positive correlation between the perceived level of competence and the perceived level of confidence when delivering CPCS (r = 0.966, *p* ≤ 0.001), indicating that the more competent the respondent felt when conducting CPCS, the more confident they felt. Respondents’ level of competence and confidence when conducting CPCS showed a small positive correlation with the number of referrals they received from NHS 111. With an increasing number of referrals, the level of competence (r = 0.259, *p* = 0.003) and confidence (r = 0.264, *p* = 0.002) increased. There was no significant correlation between the number of referrals received from GPs and confidence and competence. There were no statistically significant differences in self-reported competence and confidence according to the type of learning, pharmacist’s role, type of pharmacy, or year of qualification.

## 4. Knowledge, Confidence and Application of Skills

The learnt skills examined in the questionnaire included: Calgary-Cambridge, L(ICE)F, SBARD, clinical history-taking, clinical assessment, clinical record-keeping, and clinical transfer of care. Participants were asked to rank their level of agreement (on a scale of 1–5, 1 = strongly disagree and 5 = strongly agree) with various statements relating to their knowledge of those skills, their confidence in using those skills, and their application of those skills into the delivery of CPCS and wider practice. The distribution of responses for each statement can be seen in Figure 1. Higher levels of agreement corresponded to a better level of knowledge/confidence/application of skills.

### 4.1. Knowledge of Skills

Respondents appeared knowledgeable regarding the seven skills. The highest levels of knowledge were recorded for ‘taking a clinical history’, ‘clinically assessing a patient’, ‘using L(ICE)F’, and ‘completing an accurate and concise clinical record’, with agreement of over 80% for all these skills. A slightly lower proportion of respondents knew how to use the Calgary-Cambridge model or implement a warm transfer of care. The lowest level of agreement was found for the SBARD tool, with just under half of all respondents agreeing that they knew how to use this tool.

### 4.2. Confidence in Using Skills

Respondents also felt confident with regard to the seven skills, generally following the pattern of responses for the knowledge statements, albeit with slightly lower agreement levels. As with the knowledge statements, the lowest levels of confidence were reported for ‘warm transfer of care’ and SBARD.

### 4.3. Application of Skills in CPCS

More than half of respondents agreed that they were applying the five skills (Calgary-Cambridge, taking a clinical history, clinically assessing patients, using L(ICE)F and completing concise and accurate records during their CPCS consultations. The highest levels of agreement were found for the taking of clinical histories, clinically assessing patients, and completing concise and accurate records.

### 4.4. Application of Skills in Non-CPCS Consultations

The pattern of responses for the application of skills in non-CPCS consultations was broadly similar to that for CPCS, indicating that the learnt skills were also being applied in other pharmacy consultations.

Subgroup analysis indicated that employed pharmacists/pharmacist managers were significantly more likely than locum pharmacists to agree that they knew how to implement a warm transfer of care (70.5% vs. 48.3%, χ^2^ = 7.372, *p* = 0.007), and this group of pharmacists were also more likely to agree that they felt confident that they could enact a warm transfer of care (64.8% vs. 45.8%, χ^2^ = 5.212, *p* = 0.022). Respondents who registered as a pharmacist from 2000 onwards were more likely to agree that they were making accurate and concise clinical records in non-CPCS consultation than those who qualified pre-2000 (68.8% vs. 49.1%, χ^2^ = 5.181, *p* = 0.023). This result was not, however, significant for application into CPCS consultations. There were no significant differences according to gender or pharmacy type.

### 4.5. Summed Scores for Knowledge of, Confidence in, and Application of Skills

Responses to the statements in each of the sections on knowledge of skills, confidence in using skills, beliefs about capabilities (confidence), application of skills in the CPCS, and application of skills outside CPCS were summed together to give an overall score for each domain. Scores for the knowledge and belief domains ranged from 7 to 35, while scores for the application domains ranged from 5 to 25.

As shown in Table 2, the mean value for the knowledge statements was slightly higher than for the confidence statements, suggesting greater levels of knowledge in the skills noted than confidence. The means for ‘application of skills into CPCS delivery’ and ‘application of skills into wider practice’ were very similar. See Table 2 for details. There were no significant differences in mean values according to gender, year of qualification (pre-2000 vs. post-2000), employee type (locum vs. employed) or type of pharmacy (multiple vs. independent).

Spearman’s rank tests indicated a strong correlation between knowledge and confidence in the use of skills (r = 0.910, *p* < 0.01). Spearman’s Rank was also used to assess the correlation between ‘confidence in the use of skills’ with both ‘application of skills into CPCS delivery’ and ‘application of skills into wider practice’, with the results indicating a strong positive correlation between confidence in use and application of skills into CPCS (r = 0.811, *p* < 0.01) as well as wider practice (r = 0.756, *p* < 0.01).

## 5. Enablers and Barriers to Providing the CPCS

Respondents were asked to record their agreement or disagreement with a series of statements regarding enablers and barriers to providing the CPCS. The proportion of respondents agreeing with the statements is shown in Figure 2. The highest levels of agreement were with statements regarding the appropriateness of privacy in the consultation room for CPCS, the lack of GP CPCS referrals, patients regarding the community pharmacy as an integral part of the primary healthcare team, and that NHS 111 referrals were appropriate for handling in the community pharmacy.

In order to identify whether the statements represented an enabler or barrier to the implementation of the CPCS, negative statements (1, 2, 3, 5 and 6 in Table 3) were reverse coded, so that all results were in the same direction. Median agreement scores were calculated, having first excluded ‘not-applicable’ responses; a median score of 4 or more was defined as indicating an ‘enabler’ and a score of 3 or less was defined as indicating a possible barrier to CPCS implementation.

As shown in Table 3, a lack of referrals for CPCS from GPs (statement 1); insufficient staffing levels within community pharmacies (statement 4); difficulties in managing additional CPCS-related workload (statement 5); a lack of a positive relationship with local general practices (statement 8); a lack of awareness of the CPCS by general practices (statement 9); as well as the views of GPs and other local general practice staff members on community pharmacy not being an integral part of the primary healthcare team (statements 12 and 13, respectively), were all likely to be barriers against the implementation of CPCS.

In terms of the enablers of CPCS implementation (a median of ≥4), these included NHS 111 referring patients for CPCS (statement 2); pharmacists having a clear understanding of what is expected of them in the delivery of CPCS (statement 3); minimal concerns over indemnity cover when delivering CPCS (statement 6); adequate privacy within consultation rooms (statement 7), referrals from NHS 111 perceived as being appropriate to be dealt with in a community pharmacy setting (statement 10) and patients considering pharmacists to be integral to the primary healthcare team (statement 11). See Table 3 for details.

There were some significant differences with regard to the barriers and enablers and the type of pharmacy worked in. Those working in small/medium/large multiple pharmacies were more likely than those in independent pharmacies to report that they were not receiving GP referrals (68.5% vs. 49.0%, χ^2^ = 5.249, *p* = 0.022). Those working in independent pharmacies were more likely to agree that they had enough staff to provide the CPCS (45.3% vs. 26.3%, χ^2^ = 5.526, *p* = 0.019) and also more likely to agree that the local GP considers community pharmacy to be an integral part of the primary healthcare team (54.7% vs. 37.2%, χ^2^ = 4.214, *p* = 0.040). Employed pharmacists were more likely than locums to report that they had a good relationship with their local general practice (57.5% vs. 37.5%, χ^2^ = 5.436, *p* = 0.020). There were no other significant differences in barriers or enablers to CPCS implementation according to gender, year of qualification, pharmacy type or job role.

As outlined above: the lack of CPCS referrals from GPs; the lack of pharmacy staff; the struggle with managing CPCS-related workload; the lack of positive relationship with local general practices; the lack of awareness of CPCS by local general practices; as well as the views of GPs and other local general practice staff members on community pharmacy not being an integral part of the primary healthcare team were all identified as barriers against CPCS implementation.

Independent sample *t*-tests were used to assess the association between each of these identified barriers and the ‘application of skills into CPCS delivery’ variable (see Table 2). A higher ‘application of skills into delivery’ score indicates greater application of skills as part of CPCS consultations. The findings of the independent sample *t*-tests showed that pharmacists who felt they had sufficient staff to provide the CPCS recorded a higher ‘application of skills into CPCS delivery’ score than those who did not have enough staff (20.0 (±6.994) vs.17.5 (±5.001), t = 2.236, *p* = 0.028. In addition, pharmacists who agreed that the local GP regarded community pharmacy as integral to the primary healthcare team were more likely to record higher ‘application of skills into CPCS delivery’ scores (19.5 (±7.1421) vs. 17.2 (±4.7056), t = 2.312, *p* = 0.023). There were no other significant differences in ‘application of skills into CPCS’ score according to the other barriers.

## 6. Discussion

This study used an online survey which aimed to investigate the impact of the CPPE CPCS learning programme on pharmacist participants’ knowledge of, and confidence in, and the application of learnt skills in both CPCS and wider consultations. The study further explored barriers and enablers to providing the CPCS. The skills analysed in this study included using the Calgary-Cambridge model, taking a clinical history, clinical assessment, clinical record taking, utilising the L(ICE)F and SBARD tools, and warm transfer of care. These skills are important for providing person-centred care.

Most respondents were delivering CPCS, with the proportion of pharmacists delivering the services being similar to published figures indicating that 91% of pharmacies in England signed up to deliver the CPCS [19]. In this study, respondents reported good levels of knowledge of most skills taught in the CPPE CPCS training, but this decreased slightly when it came to confidence. Despite appearing knowledgeable and competent, respondents did not always incorporate their learnt skills to the same degree in their clinical practice (both for CPCS and other consultations). This echoes previous research, which indicates that it is not sufficient simply to acquire skills [5,14,20]. Nevertheless, it was positive to see that learning impacted more broadly on consultations other than CPCS.

Survey respondents generally had high levels of self-reported competence and confidence in delivering the CPCS, echoing previous research indicating that training for clinical services enhances confidence in service delivery [14]. There was a positive relationship between the number of referrals received and perceived competence; this is likely because higher numbers of referrals provide pharmacists with more practice opportunities, which can improve competence [8].

Pharmacists who had qualified post-2000 were more likely to report applying concise and accurate clinical records skills and conducting clinical assessments. One possible explanation for this is that pharmacists who have qualified post-2000 may be more familiar with applying clinical skills, as these skills were introduced during their undergraduate pharmacy training [21]. Also, previous research has indicated that younger professionals may be more willing to adapt to change [22]. Employed pharmacists/pharmacist managers were significantly more likely to report clinically assessing their patients than locum pharmacists. Locum pharmacists can sometimes report a lack of support for, or opportunities to, undertake advanced roles, and this, in addition to the fact that locums sometimes work shorter hours, may explain this difference [23,24].

Privacy has previously been identified as a barrier to service implementation [10] and in this study appropriate privacy was an enabler for CPCS implementation. A lack of clear service objectives has also previously been noted as a barrier to implementation [12], but in this study pharmacists appeared to have a clear understanding of what was expected from them in the CPCS. Previous studies indicated that some patients lack awareness of community pharmacy services [11,25], but in this study a significant proportion of respondents agreed that patients regarded community pharmacy as an integral part of the primary healthcare team. Pharmacists who felt competent/confident when delivering CPCS reported fewer perceived barriers to CPCS delivery, also in line with previous research [22].

One of the key barriers to the implementation of the CPCS was that pharmacists were not receiving many referrals from GP surgeries, which tallies with official figures, indicating that, as of Spring 2022, only 862 GP practices were referring patients to the CPCS service [19]. Low referrals are likely to limit the opportunities for pharmacists to apply their skills [14]. There was also evidence to suggest that GPs lacked awareness of the CPCS and that healthcare staff may not consider community pharmacy as an integral part of the healthcare team. This echoes previous research, where community pharmacists report feeling undervalued by GPs [8,26,27,28]. Collaboration between community pharmacists and GPs, important for the successful implementation of pharmacy interventions, particularly CPCS which relies on referrals, has previously been inhibited due to fears of boundary encroachment [29].

Employed pharmacists/pharmacist managers were more likely to report a good relationship with the local general practices than locums, possibly due to them having more opportunities to communicate with local GPs and build a rapport than locums, who often work in multiple locations and/or part-time [23,24]. GPs may be reluctant to collaborate with pharmacists they do not know well and need to build trust, something which takes time [27,29].

Inadequate staffing and high workload also acted as barriers to service implementation, aligning with previous research [6,30]. Measures such as employing sufficient staff or delegating responsibilities could free up the pharmacists’ time for more person-centred services, such as CPCS [31]. Pharmacists working in multiple pharmacies were more likely to report perceived barriers to CPCS implementation, in terms of inadequate staffing, low levels of GP CPCS referrals, and lack of recognition by other healthcare professionals. It has been suggested that pharmacists working in independent pharmacies are more invested in their work, cultivate better relationships with patients and better interactions with other health professionals [32] High staff turnover in multiple pharmacies can hinder the development of trust and strong relationships between GPs and such pharmacies [27].

The small sample size and low response rate are potential study limitations. There was some evidence that older pharmacists were over-represented in the study [33,34]. As the majority of participants reported that they had provided the service, yet uptake of CPCS referrals has been low, it is possible that those who had not completed any CPCS consultations may have chosen not to respond. The study also only targeted pharmacists who completed the CPPE learning, which although well attended, was not mandatory.

## 7. Conclusions

This study indicates that the CPPE CPCS learning programme positively impacted community pharmacists’ knowledge and confidence regarding specific skills relating to person-centred consultations, which the learning targeted. Whilst many respondents were applying these skills in both CPCS consultations and in wider practice, many known barriers to service implementation in community pharmacy remained. The key barrier for this particular service was a lack of GP referrals; this will need to be addressed so that referrals increase, and community pharmacists have a chance to practise their skills for improved care of patients presenting with urgent, low-acuity conditions. This study has relevance for pharmacy practice and policy, demonstrating that, with adequate training, community pharmacists can extend their practice and develop a wide range of clinical skills and contribute to enhanced provision of urgent care in England. Furthermore, this study confirms barriers, both interpersonal and infrastructural, that pharmacists encounter when trying to put their extended skills into practice, which will in turn hinder service implementation.

## Figures and Tables

**Figure 1 pharmacy-10-00117-f001:**
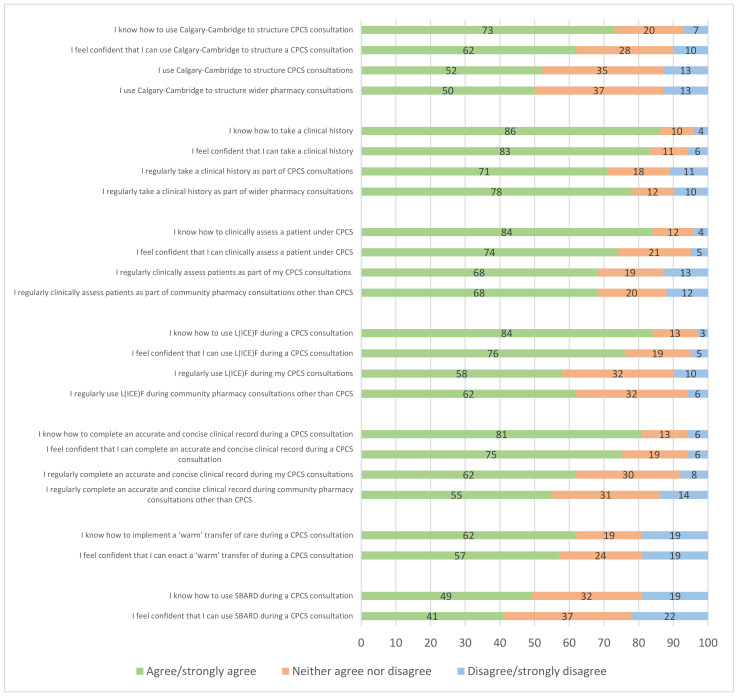
Knowledge, confidence and application of skills acquired during CPPE CPCS learning.

**Figure 2 pharmacy-10-00117-f002:**
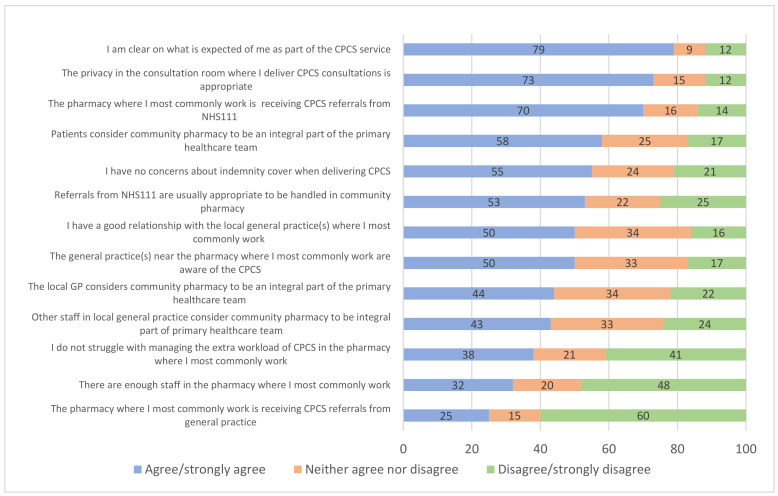
Barriers and enablers to the implementation of the CPCS *. * Negative statements have been reverse coded and reworded here, so “I have concerns about indemnity cover” now reads “I have no concerns about indemnity cover”, for example.

**Table 1 pharmacy-10-00117-t001:** Characteristics of survey respondents.

Age Groups	*n* (%)
25–34 years	14 (8.8)
35–44 years	25 (15.7)
45–54 years	44 (27.7)
55–64 years	53 (33.3)
65 years +	23 (14.5)
**Year of Registration**	
Pre-1970	2 (1.3)
1970–1979	22 (13.8)
1980–1989	48 (30.2)
1990–1999	36 (22.6)
2000–2009	30 (18.9)
2010–2015	14 (8.8)
2016+	7 (4.4)
**Ethnicity**	
Arab	3 (1.9)
Asian	62 (38.9)
Black	13 (8.2)
White	78 (49.0)
Other	1 (0.6)
Prefer not to say	2 (1.3)

**Table 2 pharmacy-10-00117-t002:** Scores for knowledge of, confidence in, and application of skills.

Domain	Mean (SD)
Knowledge of skills	26.87 (5.397)
Confidence in use of skills	25.96 (5.739)
Application of skills into CPCS delivery	17.94 (6.353)
Application of skills into wider practice	17.84 (4.816)

**Table 3 pharmacy-10-00117-t003:** Barriers and enablers to CPCS implementation.

Barrier and Enabler Statements	Median Score (Inter-Quartile Range)
The pharmacy where I most commonly work is receiving CPCS referrals from general practice	2 (3)
2. The pharmacy where I most commonly work is receiving CPCS referrals from NHS 111	4 (2)
3. I am clear on what is expected of me as part of the CPCS service	4 (1)
4. There are enough staff in the pharmacy where I most commonly work	3 (2)
5. I do not struggle with managing the extra workload of CPCS in the pharmacy where I most commonly work	3 (2)
6. I have no concerns about indemnity cover when delivering CPCS	4 (1)
7. The privacy in the consultation room where I deliver CPCS consultations is appropriate	4 (2)
8. I have a good relationship with the local general practice(s) where I most commonly work	3 (1)
9. The general practice(s) near the pharmacy where I most commonly work are aware of the CPCS	3 (1)
10. Referrals from NHS 111 are usually appropriate to be handled in community pharmacy	4 (2)
11. Patients consider community pharmacy to be an integral part of the primary healthcare team	4 (1)
12. The local GP considers community pharmacy to be an integral part of the primary healthcare team	3 (1)
13. Other staff in the local general practice(s) (e.g., nurses, receptionists) consider community pharmacy to be an integral part of the primary healthcare team	3 (1)

## Data Availability

The materials supporting the findings are not available, as survey participants did not consent to sharing this information.

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
