# Peer review of "Community Pharmacist Consultation Service: A Survey Exploring Factors Facilitating or Hindering Community Pharmacists’ Ability to Apply Learnt Skills in Practice"

_pharmacy, 2022, doi:10.3390/pharmacy10050117_

Round 1

Reviewer 1 Report

The authors provide perspective on training and barriers for non-dispensing service implementation. The article is well written and thoroughly describes the findings of the survey. The sample size and potential disproportionate responses are concerning, but are transparently referenced in the discussion. Specific notes are below:

 - Page 2, lines 45-65: CPCS appears similar to MTM in the US. Connecting those dots may help increase the international applicability of the findings. 

 - Page 2, line 70: 7000 CPCS referrals per week could use some benchmarking for those less familiar with NHS statistics. Is there a way to reference this in comparison to number of primary care office visits per week, or number of pharmacists practicing in the community setting, or some other metric to show that this is relatively high or low? 

 - Page 2, lines 77-78: Seven references are provided as background for barriers to the implementation of new services, yet the aim seeks to understand barriers. Perhaps the aim should be rephrased to identify if barriers are consistent with the literature following training, or if training mitigates some barriers? 

 - Page 3, lines 100-102: Somewhere in this section it would be helpful to understand the number of clock hours of the training (e.g., is this a 2-hour continuing education program or a multi-day, robust training)? 

 - Page 5, lines 175-178: The note about respondents receiving 399 referrals across all pharmacists. Is there a way to better benchmark that statistic vs. the 7000 referral statistic in the introduction to better understand if that is a "high volume" or "low volume" number? Perhaps that note should go in the discussion section, page 12, lines 9-11, but somehow clarifying if the CPCS volume of respondents is similar to the CPCS volume of all pharmacists would be helpful to know if the survey sample is representative of the population at large. 

 - Page 13, lines 76-78: Concluding that pharmacists without training may have comparatively less confidence is a stretch without data from a pre-post or comparison-based study.  

 - Page 13, lines 80-82: "...barriers to CPCS implementation remained..." It may be helpful to clarify that the education provided in the intervention specifically did not seem to impact CPCS implementation barriers. 

Reviewer 2 Report

The article raises the important topic of the pharmacist's role in the healthcare system and how, by using the pharmacist's professional competence more effectively, it is possible to ensure fast advice to patients and save healthcare costs.

In the Introduction section, the topic is well covered, highlighting the importance of the services provided in the pharmacy and the factors enhance and hinder these services.

Aims are clearly stated.

There was no information about the validation of the questionnaire in the materials and methods. Was it done and how? Were the study participants aware of the content of counseling tools and could adequately report their use based on this?

Regarding the results, the data is presented differently in one paragraph:

Subgroup analysis indicated that employed pharmacists/pharmacist managers were significantly more likely than locum pharmacists to agree that they knew how to implement a warm transfer of care (70.5% vs. 48.3%, =7,372, p=0.007) and this group of pharmacists were also more likely to agree that they felt confident that they could enact a warm transfer of care (64.8% vs. 45.8%, 2=5.212, p=0.022). Respondents who registered as a pharmacist from 2000 onwards were more likely to agree that they were making accurate and concise clinical records in non-CPCS consultation than those who qualified pre-2000 (68.8% vs. 49.1%, 2=5.181, p=0.023). Authors could kindly check the data.

Have the authors thought that one of the factors behind the low referral of patients to the pharmacy could have been the COVID-19 pandemic?

The authors have pointed out the low percentage and rather high age of respondents as limitations of the study, but it would be necessary to supplement, how does this affect the study results? It was rather surprising to read that middle-aged and older pharmacists, who are perhaps not always ready to go along with innovations, responded more actively.
